# What do Iranians value most when choosing a hospital? Evidence from a discrete choice experiment

**Mohammad Ranjbar[1], Mohammad Bazyar[2], Fatemeh Pahlevanshamsi[3]\*, Blake Angell[4], Yibeltal Assefa[5]**

1 Department of Health Management and Economics, Health Policy & Management Research Center, School of Public Health, Shahid Sadoughi University of Medical Sciences, Yazd, Iran, 2 Department of Health Management and Economics, Faculty of Health, Ilam University of Medical Sciences, Ilam, Iran, 3 Department of Health Management and Economics, Shahid Sadoughi University of Medical Sciences, Yazd, Iran, 4 Centre for Health Systems Science, The George Institute for Global Health, University of New South Wales Sydney, Australia, 5 School of Public Health, The University of Queensland, Brisbane, Australia

\* f.pahlevan76@gmail.com

## Abstract

**Data Availability Statement:** All relevant data are within the manuscript and its Supporting information files.

### Background

Individual preferences have preceded the use of health care services, and it has been affected by different hospital attributes. This study aimed to elicit the Iranians' preferences in choosing hospitals using a discrete choice experiment.

### Methods

A discrete choice experiment (DCE) was conducted through face to face interviews with 301 participants. The DCE was constructed by six attributes were included based on a literature review, qualitative interviews, Focus Group Discussion (FGD) and consensus development approach: waiting time, quality of care, travel time, hospital type, provider competency, and hospital facilities. individuals' preferences for hospital attributes were analyzed using a mixed logit model, and interaction terms were used to assess preference heterogeneity among individuals with different sociodemographic characteristics.

### Results

Participants had strong and significant preferences for care delivered in hospitals with 'full' (β = 0.6052, p<0.001) or 'moderate' (β = 0.5882, p<0.001) hospital equipment and with 'excellent' provider competency (β = 0.2637, p<0.001). The estimated coefficients for the "waiting time of 120 minutes" (β = −0.1625, p<0.001) and the "travel time of 30 minutes" (β = −0.1157, p<0.001) were negative and significant. The results also show that the personal characteristics such as age, education level, and income significantly affected individual preferences in choosing a hospital.

**Funding:** The author(s) received no specific funding for this work.

**Competing interests:** Authors declare that they have no competing interests.

## Conclusion

Considering people's preferences can be important given the more active role of today's patients in decision-making about their treatment processes. The results of this study should be taken into consideration by health policymakers and all stakeholders to be aware of differences in preferences of people and maximize their satisfaction. In this case, it is important to continuously involve people and consider their preferences in the design, topology, construction, and equipment of hospitals.

## Introduction

Hospitals are the most expensive components of healthcare systems [1], and account for more than two-thirds of public health resources in developing countries [2]. Public expectations of hospitals have greatly increased over recent years due to technological developments, rapid growth of medical costs, increases in non-communicable diseases leading to higher engagement with health care facilities, as well as an increase in the level of public knowledge, and the improvement of the economic and social status and other living conditions [2, 3]. Therefore, patients are now more sensitive in choosing healthcare services and centers than ever. Moreover, due to the increase in competition between hospitals, the needs and wants of patients have received more attention in recent years [3–6].

The right to choose hospitals has become an important element of healthcare systems in many countries, especially Western European countries, and has been often supported by laws [7, 8]. Furthermore, the patient's access to hospitals' information on the Internet and virtual space doubles the importance of patients' choices [9]. In such a context, the patient is considered an independent consumer and is expected to actively contribute to the choice of the hospital [10]. Understanding patient preferences can help hospitals value the most important areas for further investment to improve patient orientation, efficiency, responsiveness, and service quality [7].

Nevertheless, family physicians often greatly contribute to the process of seeking treatment and choosing a hospital to treat patients in countries that traditionally operate based on the referral system and the family physician plan [11]. It is expected that patients' preferences are also taken into consideration in this process; however, its occurrence is unclear in practice [12]. In countries such as Iran, where the referral system and the family physician plan are not fully implemented, patients and their family members often make decisions about seeking treatment and choosing a hospital. Although in some cases, due to financial problems, and lack of access to sufficient information or information asymmetries, doctors' recommendations, and opinions play an important role in patients' decisions [13], but ultimately, according to the Patient's Rights Charter in Iran, the right to choose the service provider rests with the patient [3, 8, 9, 14–16]. The way such choices are made and the factors driving them have attracted many researchers. Previous studies indicated that the choice of hospitals is affected by several factors, such as personal preferences and interests [15, 17], service cost [3, 18–24], type and severity of the disease [15, 16, 20, 25–28], waiting time to receive services [24, 29], location and distance between the place of residence and the hospital [3, 9, 15, 18–22, 24, 30–33], availability of advanced medical equipment and technology [3, 15, 24–27, 31, 34], having health insurance [18, 25–27], interpersonal skills, and employees' behavior [3, 18, 30, 35–37], physician's expertise [18, 29, 31], hospital reputation [3, 18, 19, 24, 30, 36], service quality

[3, 20, 22, 36–38], diversity and method of service provision [3, 18, 23], and patient's economic capacity [15, 20, 25–27, 39]. Several studies indicate that most people tend to choose qualified physicians and hospitals with a high reputation, regardless of the type and severity of their diseases in a context where citizens have full freedom [15].

According to previous studies, a variety of factors affect the hospitals' choice [40]. The effects of these factors are not necessarily homogeneous and vary depending on the structure, context, personal, economic, and social factors [41, 42]. Despite increasing the current knowledge, there are a number of gaps in the existing literature. Notably, most studies have investigated the importance of single factors such that the relative weight and importance of each determinant to patients and the trade-offs they would accept between them is unknown. Therefore, such studies could not simulate the choice trends that were created by changing the specific attributes of service providers. Additionally, most previous studies focused on the services of a particular center and thus may not be generalizable to a broader population. [17]. Therefore, this study aimed to elicit the Iranians' preferences in choosing hospitals using a discrete choice experiment (DCE). DCEs ask participants to indicate their preferred option in a series of hypothetical scenarios that differ based on several key attributes. It is an effective method for eliciting the stated preferences of participants [43–45] and have been widely used in health research, especially in health policy and health economics studies [45–48]. DCEs are used to reflect the real-world decision context than other methods in situations where conducting a trial or observational study is impossible or impractical [23, 44, 45]. DCEs have been used to identify and evaluate the determinants of hospital choices for patients in a number of settings [9].

## Material and methods

DCE was used to elicit public preferences for hospital selection in Yazd, Iran. DCE is a stated-preference quantitative technique, originating as a mathematical psychology method, which is designed to eliciting individuals' preferences for alternative multi-attribute commodities and services [17, 44, 49, 50]. In this technique, participants are presented with hypothetical choices between two or more alternatives that are described by a common set of attributes. Participants are asked to complete a series of such choices that comprise different levels of these attributes. It is assumed that participants select the alternative with the highest utility by considering all information provide [17, 32, 51–53]. Our DCE was undertaken in 3 sequential steps: (1) study design development, (2) data collection, and (3) data analysis.

### 1. Study design development

**1.1. Selection of attributes and levels.** Identifying and developing attributes and levels are vital for the validity of DCE [23, 50, 54–56]. Only a limited number of attributes and levels can be included in the DCE due to the complexity and to ensure the accuracy, precision, and validity of the results. Therefore, there is a need to establish a balance between the comprehensiveness of attributes and the management of participants [57].

We followed recommended guidelines for the development of attributes through a multi-stage process involving a literature review, qualitative interviews with our target population, Focus Group Discussion (FGD) and consensus development approach [15, 17, 24, 32, 37, 38, 43–45, 50, 55, 56, 58–64]. We created an initial list of 15 attributes and their levels using a literature review of published studies on stated preferences and discrete choice experiments for hospital selection in different countries. our search strategy involved searching for studies published in the last 2 decades and conducting electronic searches of relevant databases using specific terms related to hospital and DCE. These findings were used to inform face-to-face

**Table 1. Description of attributes and levels.**

| Attributes | Levels | Definition |
|---|---|---|
| Waiting time | 60 min | Waiting time was based on the time between arrivals at the hospital and getting the service |
| | 90 min | |
| | 120 min | |
| Quality of care | moderate | quality of care was described based on individual's relative understanding, knowledge, or experience of service quality |
| | good | |
| Travel time | 10 min | Travel time was described by the time taken to go to hospital from home (on private vehicles) |
| | 20 min | |
| | 30 min | |
| Hospital type | public | Hospital type was based on Hospital ownership type |
| | private | |
| | Social security | |
| Provider competency | moderate | Provider competency was based on technical skills, knowledge, ability, physician–patient communication, team care, empathy, trust, and respectful care of health care providers |
| | good | |
| | excellent | |
| Hospital facilities | poor | hospital facilities were described by the availability of all major and up to date examination or intervention equipment or drugs and Other required supplies |
| | moderate | |
| | full | |

interviews with experts. In total, by using a purposive sampling method, we conducted twenty face-to-face semi-structured interviews involving a diverse group of professionals, including hospital management experts, healthcare experts, patients, health policy makers, medical doctors, an epidemiologist, and health economists. Each interview, lasting between 1 and 1:30 hours, involved reviewing the initial list of attributes, providing detailed feedback, and ranking the attributes based on importance. Subsequently, during a two-hour focus group discussion with 9 participants from the initial interviews and research team members, 6 attributes that received the highest rankings were selected for further consideration through the consensus development approach. In the FGD, the levels related to each attribute were also selected and finalized based on the information obtained from the interviews and literature review by considering the context and features aligned with Iran's health system. Table 1 shows the attributes and levels in the final design.

**1.2. Experimental design.** Experimental design is vital for DCE studies [65]. A full factorial design creates preferences for all combinations of attributes and levels. This often leads to numerous choice tasks that make the study impractical [66]. Creating fractional factorial design, the experimental design systematically creates choice tasks that provide the best possible model estimation and estimation of main effects and possible interactions [66, 67].

In our study, 5 attributes with 3 levels, and 1 attribute with 2 levels created 486 choices ($2^1 * 3^5 = 486$) that were impossible for a survey. We used SAS software version 9.4 to create an efficient design that maximizes the D-efficiency [50, 61–64, 68]. This created a subset of the full design, containing twenty-four choice sets. In the final design, choice sets divided into 3 blocks with 8 choice sets in the Persian questionnaire (the official language of people in Iran). To familiarize participants with the DCE and test their understanding, one dominant choice set was included in each block.

Fig 1 shows an example of a choice set (translated to English). Another part of the questionnaire collected personal characteristics, including age, gender, marital status, income level, education level, and insurance status that were associated with the choice of hospital.

| Attributes | Hospital A | Hospital B |
|---|---|---|
| Waiting time | 60 min | 60 min |
| Quality of care | moderate | good |
| Travel time | 30 min | 10 min |
| Hospital type | public | Social security |
| Provider competency | good | moderate |
| Hospital facilities | poor | full |
| Which hospital would you choose? (Please tick one box only)? | ☐ | ☐ |

**Fig 1. Example of a choice set.**

To ensure the validity and reliability of DCEs in health, researchers can take several measures. These include conducting pilot tests to identify any potential issues with the design, employing cognitive interviewing techniques to evaluate participant comprehension, assessing test-retest reliability by administering the DCE on two separate occasions, analyzing internal consistency across choice tasks, establishing convergent validity by comparing outcomes with other measures, and conducting sensitivity analysis to evaluate robustness and generalizability. These measures help address bias, ensure participant engagement, and examine consistency in DCEs [69].

To confirm the final version of the questionnaire, we conducted a pilot study with 30 participants to make sure that the participants did not have any problems answering the questions or that the questionnaires were not vague and difficult for them. Based on the pilot study with 30 participants, who were not included in the main sample, we found there was no sign of respondent fatigue and task complexity, and the number of choices could be managed. As such, no changes were made to the attributes or levels of the DCE.

## 2. Data collection and sampling strategy

Following the guideline proposed by Johnson and Orme [70, 71], taking into account time constraints and other factors, we targeted a sample of 300 participants in Yazd city.

Previous studies indicated that the number of participants was large enough for reliable statistical analysis [23].

The rule of thumb formula was used to calculate the size of the study population:

$$nta/c \geq 500$$

[48]

where n represents the number of participants, $c$ is the maximum numbers of attributes in a choice scenario, $t$ is the number of choice sets and $\alpha$ indicates the number of alternatives for participants to choose from.

Of course, the threshold of 500 is the minimum threshold and it is better to consider the threshold of 1000 to increase the accuracy of the study.

Based on these calculations, our study required a minimum of 300 participants, each making 24 choices from 2 scenarios with 4 levels, exceeding the recommended threshold of 1000. We implemented cluster sampling based on comprehensive health service centers in Yazd, randomly selecting participants within each cluster and distributing paper questionnaires during face-to-face interactions. The inclusion criteria were being over 18 years old, having literacy, consent for participation in the study and cognitive ability to respond to the questionnaire.

After selecting participants and giving the necessary explanations, they were asked to carefully study each choice set and choose one of the choice tasks that they preferred the most. The process of completing each questionnaire lasted for 20 to 30 minutes. Written informed consent forms were not obtained from the participants, but they were informed that participation in the research was voluntary and completing a valid questionnaire indicated their consent. A total of 301 valid questionnaires were collected from December 10, 2022, to March 23, 2023.

## 3. Ethical statement

This research was approved by the ethics committee of Shahid Sadoughi University of Medical Sciences in Yazd based on the approval of IR.SSU.SPH.REC.1401.102. All methods were performed in accordance with the relevant guidelines and regulations. All participants voluntarily accepted our interview and were informed about all aspects of the study.

## 4. Data analysis

The random utility model provides a theoretical basis for analyzing DCE data [6]. Within this framework, it is assumed that individual $i$ should choose among $j$ alternatives and select the alternative that has the highest utility.

The utility individual $i$ derives from choosing alternative $j$ in choice set $n$ is specified as below:

$$U_{ijn} = V_{ijn} + \varepsilon_{ijn} = X_{ijn}\beta + \varepsilon_{ijn} \tag{1}$$

where $X_{ij}$ is a vector of observed attributes of alternative $j$, $\beta$ is a vector of individual specific coefficients reflecting the desirability of the attributes and $\varepsilon_{ij}$ is a random error term [6].

Different models can be developed for DCE analysis based on Eq 1.

We used the mixed logit model (MLM) in STATA$_{17}$ software to analyze data and ensure that potential heterogeneity in choices was taken into consideration. MLM assumes that the distribution εij for random coefficients is usually characterized by normal distributions. Therefore, both the preferences and the heterogeneity can be estimated in the model [6]. In this study, we first estimated the main effects in the model and then tested the interaction between the attributes and participants' characteristics to evaluate potential differences in preferences between different groups such as age, income, and education levels.

# Results

## Participants' characteristics

Table 2 reports the descriptive characteristics of the sample. A total of 275 (91.4%) out of 301 participants were female, and the largest group of participants were in the age group of 30–40 years (38.2%). About two-thirds of the participants had academic degrees. More than half of the participants had a household of less than 3 family members, and more than 61% of the

**Table 2. Participants' characteristics (n = 301).**

| Variable | | Frequency | Percentage |
|---|---|---|---|
| **Gender** | Male | 275 | 91.4 |
| | Female | 26 | 8.6 |
| **Age** | <30 | 49 | 16.3 |
| | 30–40 | 115 | 38.2 |
| | 40–50 | 85 | 28.2 |
| | 50–60 | 34 | 11.3 |
| | >60 | 15 | 5.0 |
| **Marriage** | Married | 229 | 76.1 |
| | Single | 37 | 12.3 |
| | Other | 35 | 11.6 |
| **Educational status** | Primary school or below | 7 | 2.3 |
| | Middle school | 35 | 11.6 |
| | High school | 62 | 20.6 |
| | University (academic) | 197 | 65.5 |
| **Family size** | <3 | 153 | 50.8 |
| | 3–5 | 127 | 42.2 |
| | >5 | 21 | 7.0 |
| **Insurance type** | Social security | 184 | 61.1 |
| | Health insurance | 28 | 9.3 |
| | Other | 89 | 29.6 |
| **Supplementary health insurance** | Yes | 113 | 37.5 |
| | No | 188 | 62.5 |
| **Annual income (US Dollars\*)** | < 4200$ | 294 | 97.2 |
| | > 4200$ | 7 | 2.8 |

\*1 US Dollar = 420000 IR Rials

DCE results

participants were covered by social security insurance. About two-thirds of the participants did not have any type of supplementary insurance, and most of them (97.2%) had income levels of less than 4200 US dollars per year.

Table 3 shows the results of the mixed logit model (MLM) analysis.

The results demonstrated that there were several significant predictors of participant choices in the experiment. Participants had strong and significant preferences for care delivered in hospitals with 'full' (β = 0.6052, p<0.001) or 'moderate' (β = 0.5882, p<0.001) hospital equipment and with 'excellent' provider competency (β = 0.2637, p<0.001). The estimated coefficients for the "waiting time of 120 minutes" (β = −0.1625, p<0.001) and the "travel time of 30 minutes" (β = −0.1157, p<0.001) were negative and significant. Other coefficients were not statistically significant suggesting they were not driving participant choices. The estimated SD was significant for some attributes, indicating significant heterogeneity in participants' preferences for specific attribute levels. In other words, people have different preferences based on personal differences for attributes of hospital choice.

## Heterogeneity analysis

Results of the model including interaction terms are presented in Table 4. We found that we could explain some systematic differences with personal characteristics. Based on Table 4,

**Table 3. Results of mixed logit models (main effects).**

| Attributes/levels | Mean | | SD | |
|---|---|---|---|---|
| | Coefficient | SE | Coefficient | SE |
| **Waiting time** | | | | |
| 90 min | -0.1506 | 0.1055 | -0.0161 | 0.1869 |
| 120 min | -0.1625* | 0.0785 | 0.6798* | 0.1211 |
| **Quality of care** | | | | |
| Good | 0.0929 | 0.0885 | 0.0346 | 0.1350 |
| **Travel time** | | | | |
| 20 min | 0.0326 | 0.1016 | -0.0401 | 0.1719 |
| 30 min | -0.1157* | 0.0745 | 0.6535* | 0.1672 |
| **Hospital type** | | | | |
| Private | 0.1018 | 0.0785 | 0.1326 | 0.1301 |
| Social security | -0.0502 | 0.0887 | 0.1294 | 0.2215 |
| **Provider competency** | | | | |
| Good | 0.0547 | 0.0993 | -0.0427 | 0.1763 |
| Excellent | 0.2637* | 0.1433 | 0.0432 | 0.2014 |
| **Hospital facilities** | | | | |
| Moderate | 0.5882* | 0.1761 | 0.0288 | 0.1941 |
| Full | 0.6052* | 0.1265 | -0.4792* | 0.1210 |
| **Number of participants = 301** | | | | |
| **Number of observations = 4,814** | | | | |
| **Log likelihood = -1636.1428** | | | | |
| **LR chi2(11) = 31.99** | | | | |

* Denote significance at the 0.05 level

three personal characteristics significantly affected individual preferences in choosing a hospital: Age, education level, and income. It was found that the interaction of age with waiting time was positive and significant (β = 2.3106, p<0.001). Higher values of age referred to higher levels of waiting time in the present study, indicating that older people paid more attention to longer waiting times for receiving care. The results also reported a negative interaction between high income and waiting time (β = -0.1562, p<0.001), indicating that participants with higher income were more sensitive to the waiting time to receive services. The results of Table 4 show negative and significant interactions of age (β = -0.0129, p<0.001) and income level (β = -0.0187, p<0.001) with travel time, indicating that older people and people with lower income paid more attention to travel time. Positive and significant interactions of income status (β = 0.2264, p<0.001) and education level (β = 0.1190, p<0.001) with full medical facilities indicated that higher education and income levels increased the individuals' sensitivity to choosing more equipped hospitals.

## Discussion

The present study evaluated patient preferences over different attributes of hospital care. To our knowledge, it was the first DCE that systematically elicited individual preferences over Iranian healthcare centers. We found that the choice of hospital was significantly affected by hospital facilities, provider competency, waiting time, and travel time. As expected, and based on the positive association of "provider's competency" and "hospital facilities" with participant choice, we found that hospitals with better diagnostic and treatment equipment and the higher

Table 4. Results of the preference heterogeneity analysis.

| Attributes/levels | Mean | | SD | |
|---|---|---|---|---|
| | Coefficient | SE | Coefficient | SE |
| **Waiting time** | | | | |
| **90 min** | -0.2376 | 0.2256 | -0.6210* | 0.1967 |
| **120 min** | -0.4402* | 0.4067 | -0.0224 | 0.1323 |
| **Quality of care** | | | | |
| **Good** | 0.1361 | 0.0912 | 0.6823* | 0.1371 |
| **Travel time** | | | | |
| **20 min** | 0.0879 | 0.2151 | -0.0534 | 0.1709 |
| **30 min** | -0.4190* | 0.3867 | -0.0501 | 0.1558 |
| **Hospital type** | | | | |
| **Private** | 0.0645 | 0.0782 | 0.4438* | 0.1476 |
| **Social security** | -0.0209 | 0.0897 | -0.1251 | 0.2598 |
| **Provider competency** | | | | |
| **Good** | 0.0887 | 0.1019 | 0.0775 | 0.1806 |
| **Excellent** | 0.2770* | 0.1455 | 0.0052 | 0.1857 |
| **Hospital facilities** | | | | |
| **Moderate** | 0.5665* | 0.1768 | 0.0613 | 0.2013 |
| **Full** | 0.6067* | 0.1273 | -0.3894* | 0.1452 |
| **waiting time × age** | 2.3106* | 0.1032 | | |
| **travel time × age** | -0.0129* | 0.0131 | | |
| **waiting time × income** | -0.1562* | 0.0340 | | |
| **travel time × income** | 0.0187* | 0.0455 | | |
| **full facilities ×income** | 0.2264* | 0.1013 | | |
| **full facilities × education** | 0.1190* | 0.0652 | | |
| **Number of participants = 301** | | | | |
| **Number of observations = 4,814** | | | | |
| **Log likelihood = -1538.2386** | | | | |
| **LR chi2(11) = 25.41** | | | | |

* Denote significance at the 0.05 level

competency of healthcare providers were preferred by participants while hospitals with longer waiting or travel time to receive services were less likely to be chosen by people. Other attributes, specifically the quality of medical care and the private ownership of the hospital did not significantly impact participant choices. The dominant role of hospital facilities in respondent choices was also important. These results reflected the important role of advanced technology and facilities in the selection of hospitals [16]. The availability of medicine, facilities, and technology is vital for providing high-quality care and people expect high-quality services with advanced hospital facilities [24]. Our work shows that these factors are highly valued by people. This result was consistent with previous studies [4, 16–18, 20, 23, 24, 28, 50, 55, 72, 73]. Of course, at the same time, we should also pay attention to the fact that some studies have shown that the competition of hospitals to use advanced facilities does not necessarily lead to an improvement in the quality of services or health outcomes [17, 74–76].

Our study also pointed out the important role of provider competence and indicated that it played an important role in an individual's choice preferences after hospital facilities. Previous studies also reported the significant role of "provider's competency and skill" in choosing a

hospital, and their result were consistent with the present study [16–18, 24, 31]. The provider's competency depends on the knowledge, expertise, and skill of human resources of any organization. Since human resources are considered the most important resources of healthcare organizations, their competency acts as a competitive advantage [77]. In a context, where citizens have full freedom, most people tend to choose the most competent physicians and providers, regardless of the type and severity of their diseases [15]. This is higher, especially when people have more severe diseases [17] probably due to the direct relationship between the provider's competency and the service quality. People usually expect more competent providers to provide higher-quality services [22, 35].

Our results also indicated the importance of convenience for people with travel time and waiting time significant predictors of choice. Although our study did not address this issue, the evidence indicates that waiting time and even travel time are affected by the disease severity, service quality, access to hospital, and service cost [6, 8, 16, 17]. Previous studies also indicated that when diseases were more severe, the shorter travel and waiting time to receive services were considered important [21] because patients preferred to access treatment in the shortest possible time [17]. When the cost of services was not an important challenge for patients, patients sought to choose hospitals that provided their treatment faster even at a higher cost, and thus, the shorter travel and waiting time was considered an advantage [6, 15, 16]. Patients are willing to wait in queue for more time or even travel a longer route to access better quality services when it is difficult to access the hospital or when the quality of service is important for them [20, 28, 35].

The estimates of the mixed logit model (MLM) indicated the heterogeneity of preferences among people with different demographic characteristics. Regarding the waiting time, we found that younger people and those with higher income levels were more sensitive to the waiting time and preferred hospitals that provided services in the shortest possible time. However, this sensitivity decreased at older age and even lower annual income. Elderly people and those with lower income levels were less worried about this issue. Regarding travel time, we found that elderly people and those with lower income levels were more worried about prolonged travel time, indicating their concern about travel costs, as well as their worse physical status, and unwillingness to travel long distances. The results of our research were consistent with previous studies [6, 21, 75–78].

Regarding the interaction of demographic characteristics and hospital facilities, we found a direct relationship between the preference for more equipped hospitals with the income and education levels of individuals in a way that the higher the individuals' education and income levels, the greater the individuals' desire to choose more equipped hospitals. To our knowledge, the increase in education level increases the individuals' expectation and this can increase the level of expectation from hospitals to provide higher-quality services using advanced facilities. On the other hand, the more the households' income levels are improved, the more their capacity to pay for more advanced treatments increases. The results of our study were consistent with previous studies [6, 79, 80].

The different models of preferences for different groups of participants in different conditions indicate that the government and other health policymakers should focus their efforts on key groups. Since the population of Iran is aging at a high rate, future policies of the health system in the field of access to healthcare centers should specifically consider the preferences of elderly people.

It is worth mentioning that imposing international sanctions and economic pressures against Iran, which has decreased international relations, and thus Iran's lack of access to medicine and advanced medical equipment may cause the lack of development of hospitals and lower satisfaction with hospitals. Furthermore, Iran is now experiencing an unexpected

increase in the immigration of physicians, nurses, and other elites, and it has a negative effect on the provider's competence and cause many problems for hospitals in the future if the right policies are not adopted.

Our study provides a valuable guide for the government and health policymakers to identify the key components of hospital care valued by patients in Iran. It also helps to guide appropriate decisions for better access of people to hospital services. It highlights the need to focus on vulnerable groups such as the elderly and low-income people, according to the existing conditions and challenges.

## Study limitations

As with other DCEs, our study is potentially limited by its reliance on stated rather than revealed preference data. Previous studies, which compared the results of stated preferences with actual choices, indicated that the assumed bias could be significant [81]. However, numerous studies indicated a consistency between the results of DCE and real-world decisions [44] and we followed recommended guidelines to ensure the relevance of our DCE to the real-world decision being modelled. Another type of bias is that the results of DCEs are conditioned on a limited set of included attributes. The introduction of other prominent attributes can affect the research findings. The results should be interpreted according to this issue [44].

## Conclusion

Individual choices of which hospital to receive care is more than an academic question in Iran. Individuals' choices were affected by a complex interaction between a variety of personal and hospital characteristics. Our results highlighted the relative importance of hospital facilities, provider competence, waiting time, and travel time. We also found evidence for the preference heterogeneity associated with the individuals' socioeconomic background in Iran. In particular, people's age, education level, and income caused significant differences in people's choices. Considering people's preferences can be important given the more active role of today's people in decision-making about their treatment processes. The results of this study should be taken into consideration by health policymakers and all stakeholders to be aware of differences in preferences of people and maximize their satisfaction. In this case, it is important to continuously involve people and consider their preferences in the design, topology, construction, and equipment of hospitals.

## Supporting information

**S1 Data.**
(XLSX)

## Acknowledgments

This paper has been extracted from the MSc research thesis of Shahid Sadoughi University of medical sciences. The authors would like to thank all participants who have completed the questionnaire and who helped us in gathering data and giving information to enrich the content of the study.

## Author Contributions

**Writing – review & editing:** Mohammad Ranjbar, Mohammad Bazyar, Fatemeh Pahlevan-shamsi, Blake Angell, Yibeltal Assefa.

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
