## [Decision Letter · Decision Letter 0]

6 Feb 2024

PONE-D-23-32483What do Iranians value most when choosing a hospital? Evidence from a discrete choice experimentPLOS ONE

Dear Dr. Pahlavanshamsi,

Thank you for submitting your manuscript to PLOS ONE. After careful consideration, we feel that it has merit but does not fully meet PLOS ONE’s publication criteria as it currently stands. Therefore, we invite you to submit a revised version of the manuscript that addresses the points raised during the review process.

We look forward to receiving your revised manuscript.

Kind regards,

Peivand Bastani

Academic Editor

PLOS ONE

Reviewers' comments:

Reviewer's Responses to Questions

**Comments to the Author**

1. Is the manuscript technically sound, and do the data support the conclusions?

Reviewer #1: Yes

Reviewer #2: No

2. Has the statistical analysis been performed appropriately and rigorously? 

Reviewer #1: Yes

Reviewer #2: No

3. Have the authors made all data underlying the findings in their manuscript fully available?

Reviewer #1: Yes

Reviewer #2: Yes

4. Is the manuscript presented in an intelligible fashion and written in standard English?

Reviewer #1: Yes

Reviewer #2: Yes

5. Review Comments to the Author

Reviewer #1: The authors aimed to elicit preferences of a sample of Iranian population toward choosing a hospital to receive medical care. There are some issues regarding the manuscript that should be addressed by the authors.

1- The authors stated “In countries such as Iran, where the referral system and the family physician program are not fully implemented, patients and their family members often make decisions about seeking treatment and choosing a hospital”. However, it has not been mentioned to what extent patients and their relatives have the right to choose the hospital? Basically, isn't this authority in the country of Iran, to a large extent, the monopoly of medical doctors? The effect of this important factor is not considered in the model. How have researchers managed this bias?

2- The information provided regarding the methodology of the qualitative phase of the study (literature review, qualitative study, and focus group) is incomplete. A detailed description of these steps should be provided in the main text or in the form of an additional file.

3- What measures have the researchers taken to ensure the validity and reliability of the design? Based on the model proposed by Janssen et al. (Janssen EM, Marshall DA, Hauber AB, Bridges JF. Improving the quality of discrete-choice experiments in health: how can we assess validity and reliability? Expert review of pharmacoeconomics & outcomes research. 2017;17(6):531-42.).

4- It is not clear whether the results of the pilot phase are included in the final analysis or not.

5- Despite providing information about sample size calculation, information about sampling frame, sampling method, and inclusion and exclusion criteria are completely ignored in the manuscript.

Reviewer #2: Why are the main effects of age, income and education not investigated?

Waiting time and travel time variables have three categories. Which categories of these categorical variables have an interaction effect with age and income variables?

6. PLOS authors have the option to publish the peer review history of their article (what does this mean?). If published, this will include your full peer review and any attached files.

Reviewer #1: No

Reviewer #2: **Yes: **Saeed Akhlaghi

---

## [Author Response · Author response to Decision Letter 0]

17 Mar 2024

Dear editor

I have uploaded the excel file of the raw data, re-checked the PLOS ONE's style requirements, and added ethic statements in the method section as per your suggestion. Also, I answered all the reviewers' comments as follows:

Reviewer #1: The authors aimed to elicit preferences of a sample of Iranian population toward choosing a hospital to receive medical care. There are some issues regarding the manuscript that should be addressed by the authors.

1- The authors stated “In countries such as Iran, where the referral system and the family physician program are not fully implemented, patients and their family members often make decisions about seeking treatment and choosing a hospital”. However, it has not been mentioned to what extent patients and their relatives have the right to choose the hospital? Basically, isn't this authority in the country of Iran, to a large extent, the monopoly of medical doctors? The effect of this important factor is not considered in the model. How have researchers managed this bias? 

Arthur’s answer: That is correct. In Iran, where the referral system and family physician program are not fully implemented, patients and their family members often have the freedom to decide on treatment and hospital selection. While doctors' recommendations and opinions can be influential in these decisions, patients ultimately have the right to choose their service provider.

Various factors can influence patients' choices, such as financial constraints or limited access to information about available healthcare options. In such cases, doctors' expertise and advice can provide valuable guidance for patients in making informed decisions about their healthcare.

However, it is important to note that, according to the Patient's Rights Charter in Iran, the ultimate responsibility for selecting a healthcare provider rests with the patient. They have the right to consider their own preferences, needs, and circumstances when making this decision.

According to the above text, I have corrected the relevant paragraph in the manuscript and highlighted it.

2- The information provided regarding the methodology of the qualitative phase of the study (literature review, qualitative study, and focus group) is incomplete. A detailed description of these steps should be provided in the main text or in the form of an additional file.

Arthur’s answer: As you know, DCE studies have different steps, which qualitative phase includes literature review, interview and focus group, a small part of DCE studies to identify attributes and levels. Although in the dissertation from which this manuscript is extracted, these steps are explained in Persian language in full detail and it has been made available to the researchers in Yazd University of Medical Sciences, but due to the limitation of the number of words In writing the manuscript and also maintaining the balance in writing the different parts of the manuscript, we tried to write this step as briefly as possible so that it is possible to deal with other parts of the manuscript as well. Of course, this is very common, and such a format can be seen in various articles published in this field, and you can see some of them in the sources I have introduced below:

- Kazemi‑Karyani A, Ramezani‑Doroh V, Khosravi F, Miankali ZS, Soltani S, Soofi M, et al. Eliciting preferences of patients about the quality of hospital services in the west of Iran using discrete choice experiment analysis. BMC. 2021;19:1-8.

- Tang CH, Xu J, and Zhang M. The choice and preference for public-private health care among urban residents in China: evidence from a discrete choice experiment. BMC Health Services Research (2016) 16:580 DOI 10.1186/s12913-016-1829-0

- Lendado T, Bitew CH, Elias F, Samuel S, Dawit Assele D, and Asefa M. Efect of hospital attributes on patient preference among outpatient attendants in Wolaita Zone, Southern Ethiopia: discrete choice experiment study. BMC Health Services Research (2022) 22:661 https://doi.org/10.1186/s12913-022-07874-X

- Berhane A, Enquselassie F. Patients’ preferences for attributes related to health care services at hospitals in Amhara Region, northern Ethiopia: a discrete choice experiment. Patient Preference and Adherence 2015:9

- Krinke K-S, Tangermann U, Amelung V, and Krauth CH. Public preferences for primary care provision in Germany – a discrete choice experiment. BMC Family Practice (2019) 20:80 https://doi.org/10.1186/s12875-019-0967-y

However, based on your valuable comment, I revised the qualitative phase of the study and added more information and details, and highlighted them in the manuscript.

3. What measures have the researchers taken to ensure the validity and reliability of the design? Based on the model proposed by Janssen et al. (Janssen EM, Marshall DA, Hauber AB, Bridges JF. Improving the quality of discrete-choice experiments in health: how can we assess validity and reliability? Expert review of pharmacoeconomics & outcomes research. 2017;17(6):531-42.).

Arthur’s answer: According to Janssen et al. (2017), there are several measures that researchers can take to ensure the validity and reliability of the design in discrete-choice experiments (DCEs) in health. These measures include: conducting pilot tests to identify any potential issues with the design, employing cognitive interviewing techniques to evaluate participant comprehension, assessing test-retest reliability by administering the DCE on two separate occasions, analyzing internal consistency across choice tasks, establishing convergent validity by comparing outcomes with other measures, and conducting sensitivity analysis to evaluate robustness and generalizability. These measures help ensure that DCEs produce valid and reliable results by addressing potential sources of bias, ensuring participant understanding and engagement with choice tasks, and examining consistency over time and across different metrics.

To confirm the final version of the questionnaire, we conducted a pilot study with 30 respondents to make sure that the respondents did not have any problems answering the questions or that the questionnaires were not vague and difficult for them. Based on this pilot study, there was no sign of respondent fatigue and task complexity, and the number of choices could be managed.

The text was revised and highlighted in the manuscript.

4. It is not clear whether the results of the pilot phase are included in the final analysis or not.

Arthur’s answer: The following paragraph was added to the manuscript: “the pilot study with 30 respondents, were not included in the main sample and so results of the pilot phase are not included in the final analysis”.

5- Despite providing information about sample size calculation, information about sampling frame, sampling method, and inclusion and exclusion criteria are completely ignored in the manuscript.

Arthur’s answer: information about sampling frame, sampling method, and inclusion and exclusion criteria are added and highlighted in the manuscript.

Reviewer #2: Why are the main effects of age, income and education not investigated?

Arthur’s answer: The main effects of age, income, and education have not been investigated for following reasons: 

Our main hypothesis in conducting this DCE study was that individual's choice of hospital is influenced by hospital attributes. Therefore, we used the DCE method to first identify the most important attributes of the hospital through literature review, interviews, and FGD, and then elicit the weight of each of these attributes and their levels (main effects) to contribution in choosing the hospital using the random utility model. 

As a result, we sought to estimate the weight of the principal attributes such as waiting time, quality of care, travel time, hospital type, provider competency and hospital facilities, but we found that the estimated SD in some attributes was large and significant, indicating significant heterogeneity in participants' preferences for specific attribute levels.

In other words, based on individual differences, people had diverse preferences for attributes related to hospital selection. We therefore hypothesized that individual's choices are influenced by some demographic or individual variable and used a mixed logit model to ensure that potential heterogeneity in choices was taken into account. MLM assumes that the distribution εij for random coefficients is usually characterized by normal distributions. Therefore, both the preferences and the heterogeneity could be estimated in the model. In summary, we first estimated the main effects in the model and then tested the interaction between the attributes and participants' characteristics to evaluate potential differences in preferences between different groups such as age, income, and education levels. 

2. Waiting time and travel time variables have three categories. Which categories of these categorical variables have an interaction effect with age and income variables?

Arthur’s answer: We designated the first level of each attribute as the base and compared subsequent levels with the base level to estimate the main effect of waiting time and travel time. However, in calculating heterogeneity, since we aimed to see if there is an interaction between age and income with increasing waiting time and travel time, we set the first levels of waiting time (60 minutes) and travel time (10 minutes), as the reference level. But to estimate the interaction between the waiting time and travel time variables with age and income variables, we defined the waiting time and travel time variables as continuous variables in the model.

---

## [Decision Letter · Decision Letter 1]

14 Jun 2024

PONE-D-23-32483R1What do Iranians value most when choosing a hospital? Evidence from a discrete choice experimentPLOS ONE

Dear Dr. Pahlavanshamsi,

Thank you for submitting your manuscript to PLOS ONE. After careful consideration, we feel that it has merit but does not fully meet PLOS ONE’s publication criteria as it currently stands. Therefore, we invite you to submit a revised version of the manuscript that addresses the points raised during the review process.

**Based on the reviewer`s comment and my further assessment, a major issue in this study is the patient's limited ability to choose their preferred hospital, as the doctor makes this decision entirely. This raises the question of how the researchers can accurately evaluate the patient's preferences in hospital selection. This question and concern need to be covered and justified based on the methodology and context of the study before any further decision on publishing the article.**

** **

We look forward to receiving your revised manuscript.

Kind regards,

Peivand Bastani

Academic Editor

PLOS ONE

Reviewers' comments:

Reviewer's Responses to Questions

**Comments to the Author**

1. If the authors have adequately addressed your comments raised in a previous round of review and you feel that this manuscript is now acceptable for publication, you may indicate that here to bypass the “Comments to the Author” section, enter your conflict of interest statement in the “Confidential to Editor” section, and submit your "Accept" recommendation.

Reviewer #1: (No Response)

Reviewer #3: All comments have been addressed

2. Is the manuscript technically sound, and do the data support the conclusions?

Reviewer #1: Partly

Reviewer #3: Yes

3. Has the statistical analysis been performed appropriately and rigorously? 

Reviewer #1: Yes

Reviewer #3: Yes

4. Have the authors made all data underlying the findings in their manuscript fully available?

Reviewer #1: Yes

Reviewer #3: Yes

5. Is the manuscript presented in an intelligible fashion and written in standard English?

Reviewer #1: Yes

Reviewer #3: Yes

6. Review Comments to the Author

**Reviewer #1: **Within the context of this study, a significant concern arises regarding the patient's limited ability to choose their preferred hospital. Instead, the doctor holds complete discretion in determining the hospital for medical services. In light of this, the question arises as to how we can effectively assess the patient's preferences in hospital selection.

**Reviewer #3: **(No Response)

7. PLOS authors have the option to publish the peer review history of their article (what does this mean?). If published, this will include your full peer review and any attached files.

Reviewer #1: No

Reviewer #3: No

---

## [Author Response · Author response to Decision Letter 1]

3 Jul 2024

Regarding your concern about the limited ability of patients to choose a hospital and the full competence of a physician to select a hospital, I must say that such a right does not exist for physicians in the health system of Iran. As we mentioned in our manuscript, in Iran, according to the Patient Rights Law, the patient has the freedom to choose their service provider and treatment center. On the other hand, due to the lack of a referral system in Iran, access to specialist and hospitals is freely available, and patients can choose their healthcare provider and center without any restrictions. Although the opinion of the treating physician is important, in such a context, the physician does not decide instead of the patient and does not have such a right under the law. The physician only provides advisory opinions, and the final decision to choose the treatment center is entirely the responsibility of the patient and the patient's family. It is also clear that factors such as medical costs, insurance coverage, geographical access, and similar issues can influence patient access freely. In this study, we have addressed these issues and their role in hospital selection by individuals and have shown that despite free access to hospitals, factors such as age, distance, waiting time, service quality, and other issues put patients' choice rights under scrutiny.

Another issue is that we have sought to investigate individual's preferences in this study, which hospitals they prefer to choose for treatment when they are ill. Therefore, the study results help health policymakers to take into account public preferences in redesigning the health system and building future hospitals.

---

## [Editor Report · Decision Letter 2]

6 Aug 2024

What do Iranians value most when choosing a hospital? Evidence from a discrete choice experiment

PONE-D-23-32483R2

Dear Dr. Pahlavanshamsi,

We’re pleased to inform you that your manuscript has been judged scientifically suitable for publication and will be formally accepted for publication once it meets all outstanding technical requirements.

Kind regards,

Peivand Bastani

Academic Editor

PLOS ONE
---

## [Editor Report · Acceptance letter]

12 Aug 2024

PONE-D-23-32483R2 

PLOS ONE

Dear Dr. Pahlavanshamsi, 

I'm pleased to inform you that your manuscript has been deemed suitable for publication in PLOS ONE. Congratulations! Your manuscript is now being handed over to our production team.

Kind regards, 

on behalf of

Dr Peivand Bastani 

Academic Editor

PLOS ONE